# Experimental Investigation on Jet Vector Deflection Jumping Phenomenon of Coanda Effect Nozzle

**Shaoqing Chi** and **Yunsong Gu** *

Key Laboratory of Unsteady Aerodynamics and Flow Control, Ministry of Industry and Information Technology, Nanjing University of Aeronautics and Astronautics, Nanjing 210016, China; shaoqing_chi@sina.com
* Correspondence: yunsonggu@nuaa.edu.cn

**Abstract:** The Coanda effect nozzle is a fluid thrust vectoring technology that uses the Coanda effect to control jet vector deflection. The jumping phenomenon often occurs in the process of controlling jet vector deflection. This phenomenon leads to the nonlinearity of thrust vector control. It destroys the control performance of the aircraft and brings potential dangers to the safety of the aircraft. The jumping phenomenon occurs in an unsteady flow field different from the traditional flow phenomenon. The flow structure in an unsteady flow field changes with time, so it is not easy to control by the traditional active flow control method. This paper explains the reasons for the jumping phenomenon from two aspects: flow field stability and flow structure. Secondly, the unsteady flow field with the jumping phenomenon is studied and analyzed by a flow visualization experiment and dynamic force measurement. Furthermore, the dynamic modal decomposition (DMD) method is used to extract the characteristic frequencies of the critical vortices causing jets to jump in unsteady flow fields. Finally, a pulsed jet with the same characteristic frequency is used to control the varying vortices in the unsteady flow field. The experimental results show that the active flow control method, which extracts the characteristic frequency of the critical flow field structure by DMD, effectively suppresses the jumping phenomenon in the unsteady flow field. It also linearizes the process of jet nonlinear vector deflection.

**Keywords:** Coanda effect; fluid thrust vectoring nozzle; active flow control; the jumping phenomenon; DMD

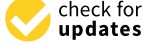



## 1. Introduction

The jumping phenomenon of jet vector deflection is one of the critical problems of fluid thrust vectoring technology. The jumping phenomenon commonly exists in various forms of fluid thrust vectoring technology. The sudden jump problem will cause the aircraft's attitude to change suddenly even if the pilot does not control it, which can very easily cause flight accidents. Therefore, the jump problem is the bottleneck and obstacle for the development and engineering application of passive thrust vector control technology based on the Coanda effect.

The Coanda effect control method is derived from the aerial Coanda high−efficiency orienting−jet nozzle (ACHEON) program [1–4] funded by the European Union in 2013. The program proposes to improve the deflection efficiency of the vectoring nozzle by using the Coanda effect of high−speed jet on the convex surface and the effect of plasma−accelerating fluid delay separation. This control method adds a trailing edge plate at the outlet of the nozzle and uses the wall−attached effect of the fluid itself to control the deflection vector. This control method is also called trailing edge plate control. The sensitivity of plasma control to flow control is also significantly improved through this method. In 2016, Lu [5] et al. studied the specific application of the plasma control method, and further elaborated on the application scope of an ion exciter in flow separation control. In the same year, Michel et al. [6–8] tried to produce a practical Coanda effect nozzle design guide, which was designed to meet the actual needs of different projects. Michel et al. established the

mathematical model of the Coanda effect nozzle based on the ACHEON project in the form of the integral equation, proposed a new aircraft architecture based on this model, and verified the model based on this nozzle through numerical simulation.

In 1960, Newman [9] proposed the inclined wall jet model, and suggested that the dimensionless reattachment distance is only a function of the wall deflection angle. Subsequently, the model has been widely studied. The position of the inclined wall and the outlet can be divided into two cases: no potential difference and potential difference. The main structural characteristics of the flow model are that the jet attaches at a certain distance downstream of the outlet and that there is a recirculation zone upstream of the reattachment point, with separation bubbles, reattachment points, and other flow structures. In 2000, Lai et al. [10] obtained consistent results of reattachment positions using pressure measurement and flow display, pointing out that the reattachment positions increase with the increase of wall deflection angle.

In 2013, Asghar, et al. [11] of the Iranian university of science and technology studied the vortex position, size, velocity, average turbulence intensity, Reynolds stress, and reattachment length (attachment point) in the recirculation zone under different wall inclination angles, with a potential difference using experiments and calculations, and obtained the near−wall velocity field and pressure field in the steady−state.

In 2014, Shantanu et al. [12] used a numerical simulation method to study the turbulent wall jet flow field structure under different deflection angles, including downstream velocity field, Reynolds stress, and wall static pressure distribution. The influence of the deflection angle on the flow field structure was studied in the range of small deflection angles. The above literature research results show that the main near−wall flow structure characteristics of the Coanda flow model in steady−state wall attachment include the separation bubble and reattachment structure. For the wall−attached deflection jet's flow structure, scholars' research mainly focuses on the near−wall flow structure and flow characteristics when the wall is stable. The transverse pressure gradient between the space environment and the separation area determines jet detachment and attachment. Zaitsev et al. [13] studied the flow characteristics of a supersonic jet on the inclined wall with a fixed deflection angle and, for the first time, obtained the pressure in the starting area, the range of the recirculation area, the jet trajectory, and pointed out that the jet separation and attachment are determined by the transverse pressure gradient developed between the space environment and the separation area. Allery et al. [14] used a hot wire anemometer to conduct experimental research on inclined wall jets with different incoming velocities at a fixed deflection angle of 30 degrees. They gave the deflection angle and Reynolds number boundary conditions for jet attachment and separation.

In 2015, Gillgrist et al. [15] used particle image velocimetry and conventional pressure sensors to obtain the results of the transient velocity field. The Reynolds stress field of the near−wall flow under the stable control state for the reverse flow vectoring nozzle model demonstrated that the lateral pressure gradient generated by the negative suction pressure and the reverse flow shear layer, under the combined action of the induced negative pressure on the wall, is the cause of the jet vector deflection. Allery et al. [16] experimentally conducted numerical simulation research on the phenomenon of wall−attached deflected jet due to the Coanda effect. Through practical means, they studied the effects of the inclination angle and Reynolds number on the wall−attached and separation phenomena, as well as hysteresis and jumping. They used the Galerkin projection of the Navier–Stokes equation on POD basis function to obtain a low−dimensional dynamic model, which qualitatively represented flow characteristics. Miozzi et al. [17] studied the phenomenon of the jet deviating from the straight direction due to the presence of the Coanda wall from the experimental point of view. The velocity field results clearly show that the inclination of the jet attached to the wall depends on the side wall distance itself. The self−similarity analysis along the inclined jet direction shows that the Coanda wall attachment effect will fail for wall distances more significant than five jet widths. Cornelius et al. [18] studied the physical mechanism of the Coanda jet wall separation process under a high−pressure ratio

and qualitatively described the physical characteristics of wall−attached expansion and jet separation using an optical schlieren system.

For complex dynamic systems, commonly used dimensionality reduction methods include the inertial manifold/approximate inertial manifold method, the Eigen orthogonal decomposition POD method, and the primary manifold method [19], in which the POD method is widely used in the dimensionality reduction modeling of fluid dynamic systems. Because the POD method has a relatively small calculation, the numerical results or experimental data based on high−resolutions have clear physical significance. Lumley [20,21] first introduced the POD method into the turbulent field, and then Sirovich [22] introduced the snapshot method to study the dynamics of fluctuating flow and the dynamics of wave flow. Deane [23] et al. and Cao et al. [24] conducted a numerical simulation of the flow around a cylinder through the POD Galerkin method, and found that the evolution characteristics of the flow field could be accurately captured by using a less pod basis. The fluid thrust vectoring nozzle was an unsteady flow process in the process of vector deflection. As far as the usual structure analysis method of unsteady flow is concerned, the dynamic mode decomposition (DMD) method is commonly used. Schmid developed the DMD method from Koop in recent years; it is a low−dimensional system decomposition technology developed based on man analysis [25]. Dynamic mode decomposition (DMD) can solve or approximate the dynamic system according to the coherent growth structure, attenuation, and oscillation in time. This method has been widely used in the study of various unsteady flows, and derived the forms of optimal (opt−DMD) [26], optimal mode decomposition (OMD), and sparse improved DMD (SPDMD) [27], which has gradually became a new tool for hydrodynamics mechanism analysis. Using POD and DMD techniques, Sajadmanesh et al. [28,29] successfully identified separated bubbles in ultra−high lift turbine cascades and the flapping phenomenon in highly loaded low−pressure turbine cascades.

This paper uses the practical active flow control method to control the jet's transient vector deflection jumping phenomenon. The specific practical steps are as follows: firstly, the PIV experiment, oil flow experiment, dynamic pressure measurement, and dynamic force measurement experiment determine the control object and control position required by the active flow control method; secondly, the dynamic mode decomposition technique is used to determine the control frequency of the pulsed jet; finally, an experimental technique is used for verifying the effect of the active flow control experimental technique.

## 2. Experimental Setup and Methods

### 2.1. Coanda Effect Nozzle

The structure and mechanism of the Coanda effect nozzle are shown in Figure 1. The nozzle was comprised of a pair of Coanda walls with inclination, a secondary flow passage, a secondary flow control valve, and a central jet flow passage. As shown in Figure 1a, the secondary flow control valves at the upper and lower sides remained open when the jet remained neutral. At this time, two secondary flow fields were located at the upper and lower sides of the main jet and in the same direction as the main jet, respectively. The secondary flow was generated by the ejection effect of the main jet. This secondary flow was passive and did not need external energy injection. As shown in Figure 1b, when the jet deflected downward to the attached wall, the secondary flow control valve on the upper side of the nozzle remained open, and the passive secondary flow control valve on the lower side of the nozzle was closed. The Coanda effect was generated between the main jet and the lower side wall. At the same time, the secondary flow on the upper side of the jet generated a downward pressure difference, which eventually caused the jet to attach to the Coanda wall and complete the deflection attachment.

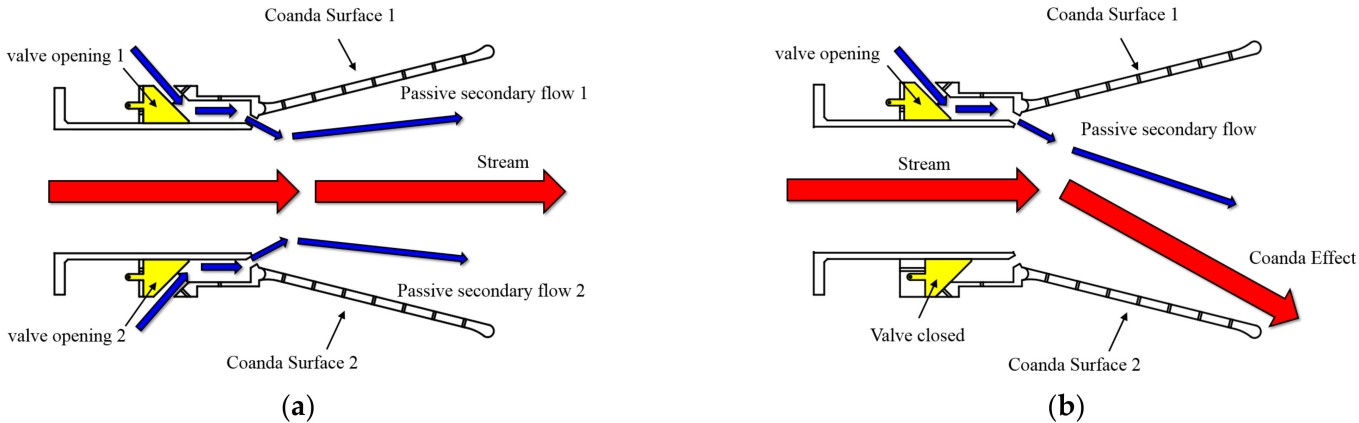

**Figure 1.** Structure diagram of passive thrust vector. (**a**) Flow structure diagram of a jet under detachment condition; (**b**) flow structure diagram of jet deflection wall attachment condition.

### 2.2. Model Parameters and Experimental Setup

The Coanda effect nozzle is shown in Figure 2. The height of the main jet outlet was D = 40 mm, and the width was L = 100 mm. There ere passive secondary flow control joints with height h = 4 mm and width W = 100 mm, respectively, at the upper and lower parts of the main jet outlet area. The Coanda wall had an 18° deflection angle with the main jet direction, the plate length L was 100 mm, and the size of the whole plate was 100 mm × 100 mm. In this article, the fluid medium was air, and the incoming flow velocity was 30 m/s. At the upper and lower walls of the nozzle, four rows of pressure taps were arranged at S1 = 1.25D, S2 = 1D, S3 = 0.5D, and S4 = 0.1D, respectively. The pressure taps were connected with the dynamic pressure sensor to measure the dynamic pressure parameters of the wall. The vectoring nozzle controlled the injection flow of passive secondary flow by adjusting the opening of the control slot. The purpose of vector control was achieved by actively controlling the pulsed jet to control the main jet's flow.

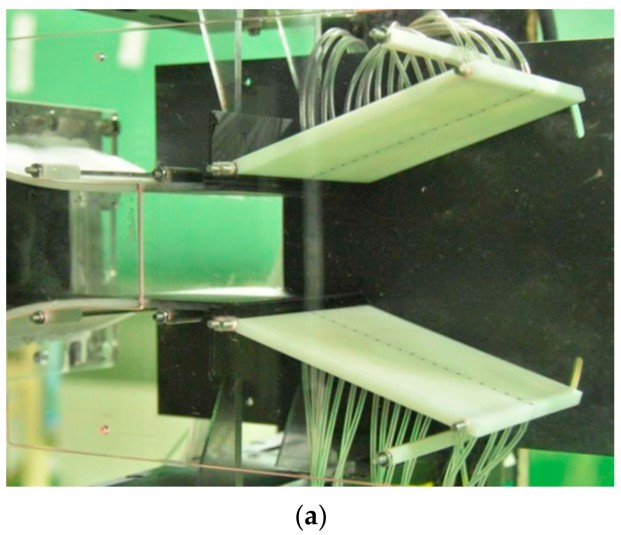

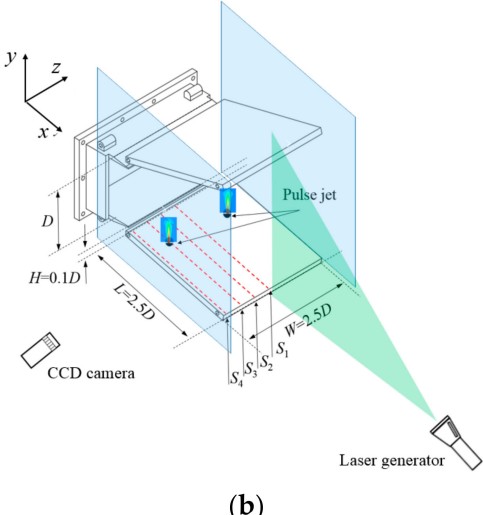

**Figure 2.** The Coanda effect nozzle model and structural dimension schematics. (**a**) The Coanda effect nozzle model in PIV Layout; (**b**) structural dimension of model and section diagram of PIV shot.

The pulsed jet convection field was used for active flow control to improve the jet deflection rate under the condition of the jet attached to the wall. The structure of the pulsed jet generator is shown in Figure 3a. The pulsed jet exciter used the active compressed air source as the energy input of the pulsed jet. The pressure reducing and stabilizing valve was used to ensure the speed stability of the output jet. The signal generator was used to generate the frequency signal. The signal amplifier was used to input the frequency signal

into the high−frequency solenoid valve. Finally, the high−frequency electromagnetic valve controlled the jet to output the constant frequency pulse jet. Among them, the model of a high−frequency solenoid valve was ZTO−45A−AA1−DDFA−1BA, which was driven by a 24VDC power supply, the maximum control frequency was 233 Hz, the power on response time was 6 ms, and the power off response time was 2 ms.

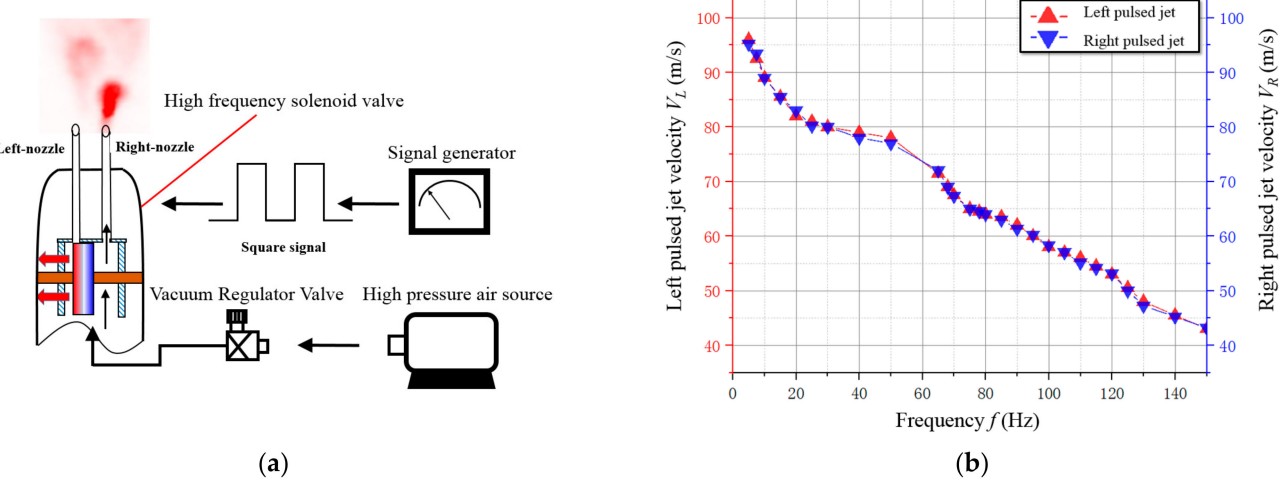

(**a**)      (**b**)

**Figure 3.** The pulsed jet exciter. (**a**) Structure diagram of pulsed jet generator; (**b**) frequency characteristic curve of the pulsed jet generator.

The pulsed jet adopted an active air source, and inputted a square−wave signal with a duty cycle of 50% through the signal generator to control the high−frequency solenoid valve. By adjusting the frequency of the square wave signal, two jet holes could release a pulsed jet with a specific frequency. The diameter of the two pulsed jet holes was d = 2.5 mm, and the spacing was 1.5D = 60 mm. The curve of the velocity of pulsed jet, varying with frequency, is shown in Figure 3b. The velocity of the jet of the active source was V = 100 m/s, and the velocity of the pulsed jet decreased with the increase of the control frequency. The momentum ratio of the pulsed jet to the main jet was 1.47–0.736%. The location of the jet hole was the lowest CP coefficient points (Z/W = 0.2, X/L = 0.215) and (Z/W = 0.8, X/L = 0.215) when the jet was in attachment condition.

Figure 4 shows the schematic diagram of the jet wind tunnel platform. In the experiment, the charge−coupled device (CCD) camera was used to record the flow field data, and its resolution was 2048 × 2048 pixels$^2$. The model of the CCD camera was Fast−CAM Mini ax50, and the camera's resolution was 1024 × 1024. The sensor adopted a CMOS sensor. The frame rate could reach 2000 Hz, and the minimum exposure time was 1.05 μs. The dynamic flow field structure in jet deflection and wall attachment was qualitatively displayed. The physical parameters in the dynamic flow field were measured. The passive secondary flow fluid thrust vectoring nozzle model was connected with the jet wind tunnel through the transition section. In the experiment, an Nd: YAG 200−MJ laser with a wavelength of 532 nm was used to illuminate the convection field. The shooting plane was perpendicular to the horizontal plane in the PIV experiment, and was located at Z/W = 0.5. The experiment performed at a velocity of 30 m/s and a free stream turbulence intensity below 0.3%. In the experiment, the dynamic pressure measurement was synchronized with the PIV experiment, and the dynamic force measurement experiment was carried out separately.

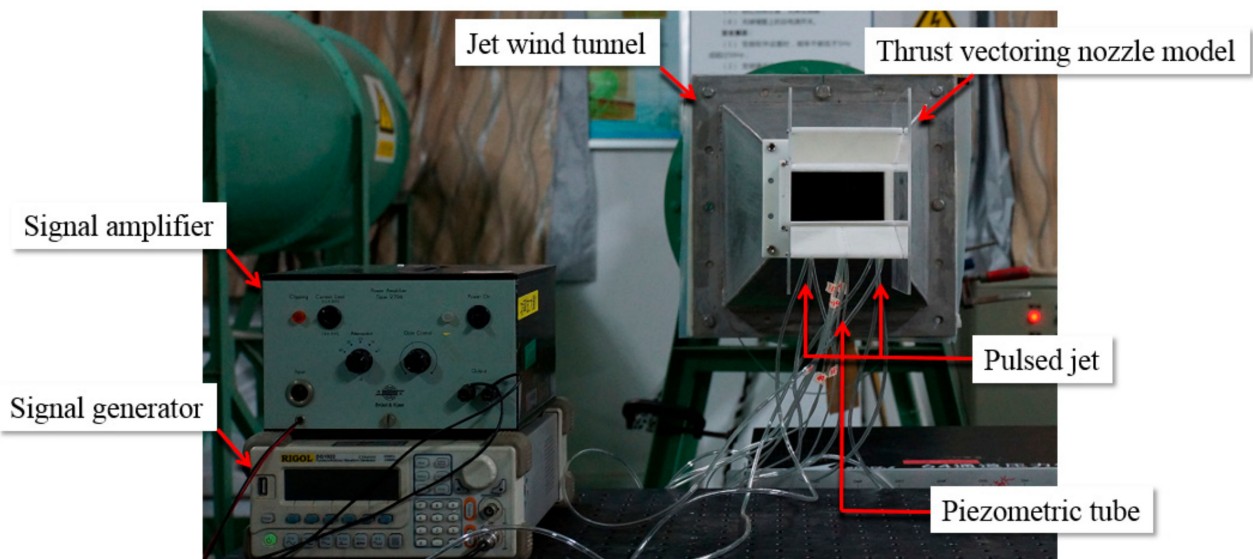

**Figure 4.** The layout of the dynamic pressure measurement on the active flow control experiment.

As shown in Figure 5a, an eight−channel pressure sensor module was used for dynamic pressure acquisition. The sensor used was SM5652−001−D−3−SR, compensated and calibrated by ceramic DIP under constant pressure excitation. Eight 8−channel pressure sensors with 64 channels and a pressure measuring the frequency of 1000 Hz were used in the experiment. Table 1 shows the technical parameters of the dynamic pressure sensor module. As shown in Figure 5b, the ATI−9610 six−component strain gauge balance made by ATI Corporation, USA, consisted of floating and fixed frames. The floating frame was fixed to the nozzle model, the bottom of the fixed frame was connected to the base, and the force, moment range, and total range measurement accuracy of the balance axes X, Y, and Z are shown in Table 2. The strain balance converted the acquired force signal into an electrical signal, amplified it through the back−end amplifier, and transmitted it to the NI data acquisition card for acquisition and storage. The acquisition frequency of ATI force balance was 200 Hz.

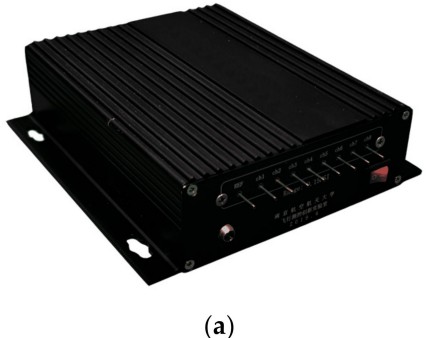

(**a**)

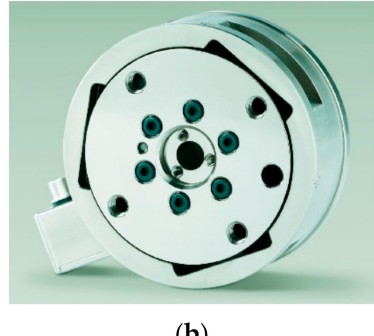

(**b**)

**Figure 5.** Dynamic pressure and force measuring test equipment. (**a**) Eight−channel pressure sensor module; (**b**) the ATI−9610 six−component strain gauge balance.

**Table 1.** Technical parameters of the dynamic pressure sensor module.

| Parameter | Numerical Value |
| --- | --- |
| Pressure measurement range | 0.15 PSI |
| Acquisition accuracy | 0.1 Pa |
| Acquisition resolution | 0.3 Pa |

**Table 2.** Measuring range and accuracy of each measuring axis of ATI balance.

| Variable | $F_x$ | $F_y$ | $F_z$ | $M_x$ | $M_y$ | $M_z$ |
|---|---|---|---|---|---|---|
| Range | 165 N | 165 N | 495 N | 15 N·m | 15 N·m | 15 N·m |
| Precision (FS) | 1.00% | 1.00% | 1.00% | 1.00% | 1.00% | 1.00% |

The dynamic pressure measurement baseline results are shown in Figure 6a. The random error of the measured pressure value within 1800 ms met the instrument standard. Furthermore, the pressure fluctuation had no evident periodicity within 1800 ms, so the pressure measurement error did not affect the experimental results. The dynamic torque measurement baseline results are shown in Figure 6b. The random error of the measured torque value within 700 ms met the instrument standard. Furthermore, the pressure fluctuation had no evident periodicity within 700 ms, so the dynamic torque measurement error did not affect the experimental results.

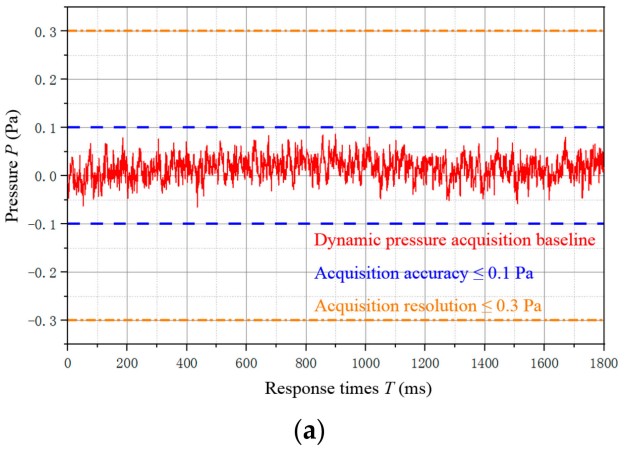

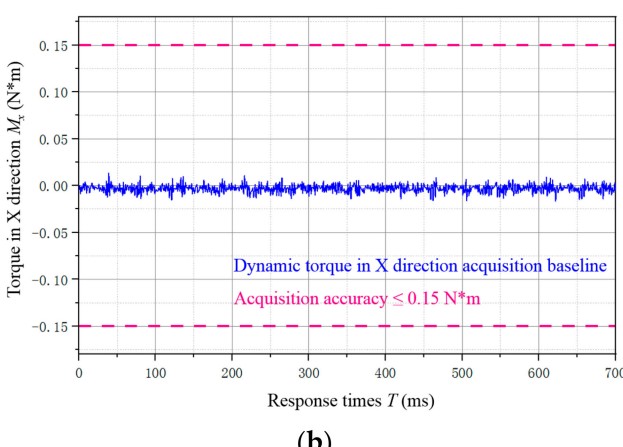

(**a**)　　　　　　　　　　　　　(**b**)

**Figure 6.** The actual error results of dynamic pressure and force measurement experiments. (**a**) The dynamic pressure measurement baseline results; (**b**) the dynamic torque measurement baseline results.

The calculation formula of the force vector angle was:

$$\theta_E = \arctan(F_X / F_Y) \tag{1}$$

$$\theta = \theta_E - \theta_0 \tag{2}$$

The force measurement results selected the forces $F_X$ and $F_Y$ in $X$ and $Y$ directions and calculated the thrust vector angle under various working conditions through Formula (1) $\theta_C$. Zero vector angle $\theta_0$ was the thrust vector angle measured by the force balance when the jet remained horizontal. The thrust vector angle obtained through the measurement and calculation of each experimental state $\theta_E$ deducted the reference zero value of vector angle $\theta_0$ to get the actual thrust vector deflection angle of the jet $\theta$. Formulas (1) and (2) effectively eliminated the error caused by assembly with the model.

*2.3. Proper Orthogonal Decomposition*

A group of transient information of flow field $\{u_1, u_2, \ldots, u_N\}$ was described as:

$$u_i = \frac{1}{N} \sum_{j=1}^{N} u_j + v_i \tag{3}$$

where $u_i$ was the transient flow field variable at time $i$, and $v_i$ was the pulsation after subtracting the average value. The POD method used a linear combination of a set of optimal orthogonal basis functions to represent $v_i$ [19], expressed as:

$$v_i = \sum_{j=1}^{N} a_j(t_i) p_j \tag{4}$$

where $p_j$ was the modal basis function of POD and $a_j(t_i)$ was the modal coefficient of mode $p_j$ corresponding to time $t_i$. The definition matrix $C = V^T V$, where $V = \{v_1, v_2, \dots, v_N\}$. Then, the eigenvalues were solved as:

$$CA^j = \lambda_j A^j \tag{5}$$

From Equation (5): $\lambda_j$ was the eigenvalue; $A^j$ was the corresponding eigenvector matrix, namely the modal coefficient matrix [20,21]; and $A^j = [a_j(t_1), a_j(t_2), \dots, a_j(t_N)]^T$ rearranged the eigenvalues by size. The 1 order POD mode corresponded to the maximum eigenvalue. The pod mode could be solved in one step:

$$p_j = \frac{1}{N\lambda_j} \sum_{i=1}^{N} A_i^j v_i \tag{6}$$

POD could calculate the energy level of each mode in the flow field [22]. Energy was defined as:

$$E_i = \lambda_i / \sum_{j=1}^{N} \lambda_j \tag{7}$$

*2.4. Dynamic Mode Decomposition*

For the flow field change information $\{u_1, u_2, \dots, u_N\}$, it was assumed that there was a matrix $A$, so that there was a linear transformation relationship between adjacent time layers:

$$u_{i+1} = Au_i \tag{8}$$

Definitions $\psi_0 = \{u_1, u_2, \dots, u_{N-1}\}$ and $\psi_1 = \{u_2, u_3, \dots, u_N\}$ could give the following relation:

$$\psi_1 = A\psi_0 = [Au_1, Au_2, \dots, Au_{N-1}] \tag{9}$$

For finding matrix $A$, DMD used a low−dimensional optimal approximation matrix $\tilde{A}$ to replace $A$. Solving $\tilde{A}$ required a singular value decomposition of $\psi_0$:

$$\psi_0 = U\Sigma W^H \tag{10}$$

where $U$ was a left orthogonal matrix, $\Sigma$ was a singular value diagonal matrix, $W$ was a right orthogonal matrix, and H was a complex conjugate transpose [25]. The approximation matrix could be expressed as:

$$\tilde{A} = U^H \psi_1 W \Sigma^{-1} \tag{11}$$

The next step involved finding the eigenvalue $\tilde{A}\Lambda_j = \lambda_j \Lambda_j$ for $\tilde{A}$, where $\Lambda_j$ was the eigenvector corresponding to the eigenvalue $\lambda_j$. Where DMD mode was $\Phi_j = U\Lambda_j$, the mode amplitude was $\alpha_j = \Lambda_j^{-1} U^H u_1$, and the mode growth rate was $g_j = \text{Re}[\ln\lambda_j / \Delta t]$. The reconstructed flow field could be expressed as:

$$u_j \approx \sum_{i=1}^{N} \phi_i (\lambda_i)^{j-1} \alpha_i \tag{12}$$

where $(\lambda_i)^{j-1} \alpha_i$ was the modal coefficient.

The residual corresponding to the maximum difference between the sample data matrix $A$ and the reconstructed data matrix $A'$ was:

$$R = \max_{\substack{1 \le i \le n \\ 1 \le j \le m}} \left| A_{i,j} - A'_{i,j} \right| \tag{13}$$

In which $A_{i,j}$ and $A'_{i,j}$ were the elements of the $A$ and $A'$ matrices, respectively [28,29].

## 3. Results and Discussion

### 3.1. Jumping Phenomenon

Figure 7a shows the measured values at diffusion angle $\alpha = 12\text{–}22^\circ$ in the experiment, with the jet vector deflection angle $\theta$ and the secondary flow valve opening $\delta$ change curve. It is defined in the figure that when the jet is in the neutral condition, the jet vector deflection angle $\theta = 0^\circ$; when the jet is attached to the wall, $\theta = \alpha$. Defining the secondary flow valve opening as $\delta = 0$, the secondary flow valve is fully opened, and the jet maintains a neutral working condition; when the opening of the secondary flow valve is 1, the secondary flow valve is completely closed, and the jet is attached to the Coanda wall at the closing side.

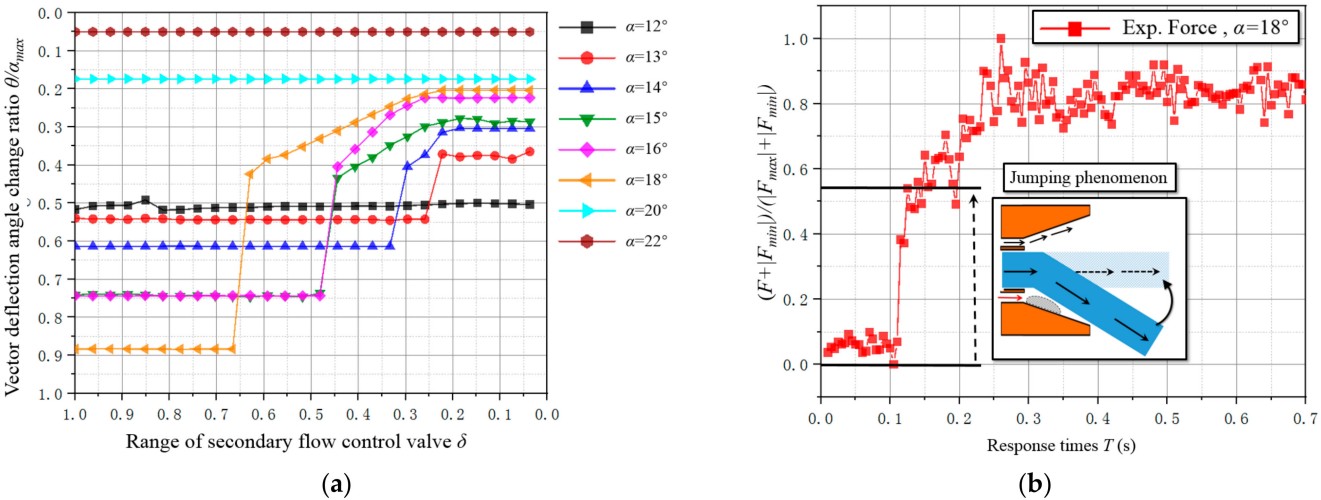

(a) (b)

**Figure 7.** The jumping phenomenon of jet vector deflection in the Coanda effect nozzle. (**a**) Jet vector deflection angle $\theta$ with secondary flow valve opening $\delta$ variation curve (diffusion angle $\alpha = 12\text{–}22^\circ$); (**b**) force variation curve of vectoring nozzle during jet dynamic detachment (diffusion angle $\alpha = 18^\circ$).

On the condition of a diffusion angle of $\alpha = 12^\circ$, the vectoring jet attaching to the wall is stable. This phenomenon shows that when the diffusion angle of the vectoring nozzle is too small, the Coanda effect will cause the jet to attach to the wall. When the diffusion angle is $\alpha = 13\text{–}18^\circ$, there is an evident jumping phenomenon in the change curve of the jet vector angle, and the vector angle does not have a linear relationship with the valve opening of secondary flow. When the secondary flow valve is gradually opened, the jet will remain attached to the wall for a while until the secondary flow valve is opened to a critical opening, and the jet will then suddenly leave the wall, resulting in a sudden jump. The vector change angle of the jumping phenomenon, $\Delta\theta$, with the diffusion angle within a specific range, $\alpha$, gradually increases. When $\alpha > 20^\circ$, the jet remains neutral and will not produce vector deflection when the Coanda effect occurs. Therefore, in the Coanda effect nozzle, the Coanda effect has a specific effective range $\theta_C$. When the distance between the vector jet and Coanda wall $\le \theta_C$, the jet will have the Coanda effect with the wall, resulting in the jet jumping to the Coanda wall; when the vector jet is attached to the Coanda wall, sufficient secondary flow will make the vector jet leave the Coanda wall and jump at the vector angle $\Delta\theta = \theta_C$.

Due to the spatial distance between the jet and the wall when the jet deflects towards the wall, it is not easy to install the pulsed jet exciter. Therefore, the phenomenon of the jet jumping off of the wall is studied in this paper. In order to better study the jump phenomenon in the Coanda effect nozzle, the expansion angle with the most apparent jump phenomenon is selected ($\alpha = 18°$) as the working condition and is taken as the research object. Figure 7b shows the expansion angle and time−dependent force curve of the vectoring nozzle during jet dynamic wall separation at $\alpha = 18°$. It can be seen from the figure that the nozzle will jump instantaneously after the force lags for a while, until the secondary flow reaches a certain level.

### 3.2. Analysis of the Attachment Flow Structure

The PIV image at Z/W = 0.5 inside the Coanda effect fluid thrust vectoring nozzle is shown in Figures 8 and 9. The region between the jet and the Coanda wall is region A. The velocity nephogram of the jet under neutral working conditions is shown in Figure 9a. Under this working condition, the secondary flow control valves open between the upper and lower sides of the jet. As shown in Figure 9b, when the main jet flow deflects downward, the upper control valve is open, and the lower control valve is closed. Only one passive secondary flow is injected in the same direction as the main jet into the nozzle flow field. The main jet remains separated from the side of the injection passive secondary flow and attached to the side of the closed valve. When the main jet flows to the lower wall, the internal flow field structure of the nozzle is divided into the main jet, the shear layer outside of the main jet, the separation bubble, and the return basin. The velocity difference between the main jet and the flow field in the nozzle causes the shear layer, and the shear layer leads to a large velocity gradient in the nozzle. The *K–H* instability caused by the high−speed shear sucks up many small vortex structures. There are apparent separation bubble structures on the wall side of the main jet, and several small and stable vortex structures in the separation bubble.

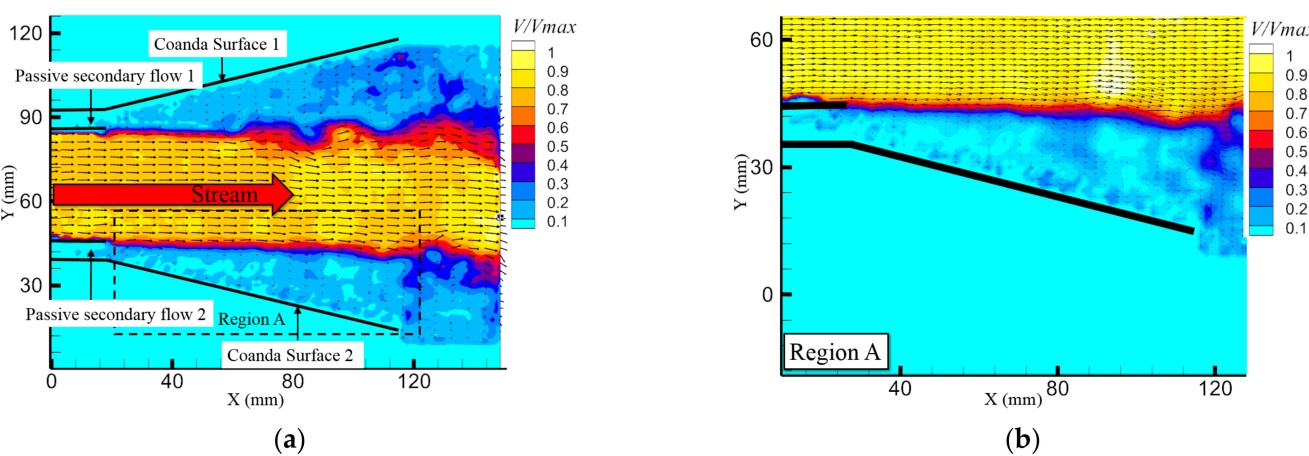

**Figure 8.** Detachment condition PIV visualization. (**a**) Velocity vector and velocity nephogram of the whole flow field under the jet detachment condition; (**b**) velocity vector and velocity nephogram of the Region A under the jet detachment condition.

In the POD method, the flow field is sorted according to the energy series, and the first−order mode is generally regarded as the time−average flow result of the flow field. The PIV results of jet attachment and jet detachment are usable to construct the snapshot set (snapshot = 100). As shown in Figure 10a, the first−order energy level in the wall−attached state is lower than in the neutral state. The first−order modal energy in the wall−attached state is 48.08%, while the first−order modal energy in the neutral state is 71.506%. It can be seen that the stability of the jet in a neutral state is higher than that in an attachment state. Therefore, when the jet deflects to the wall, the jumping phenomenon will occur after

the jet vector deflects a certain angle, and when the jet deflects away from the wall, the jumping phenomenon will occur suddenly.

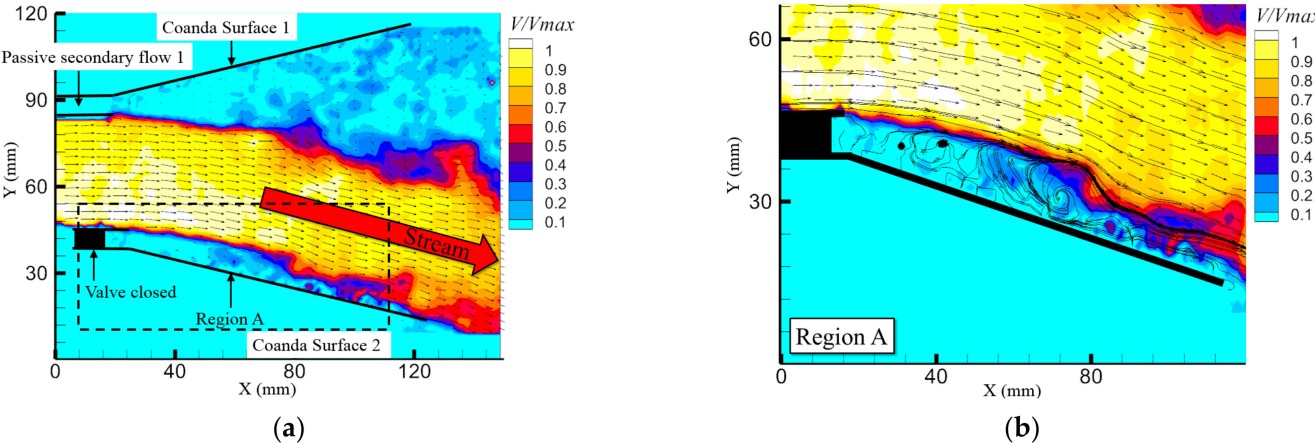

**Figure 9.** Attachment condition PIV visualization. (**a**) Velocity vector and velocity nephogram of the whole flow field under the jet attachment condition; (**b**) velocity vector, velocity streamline and velocity nephogram of the Region A under the jet attachment condition.

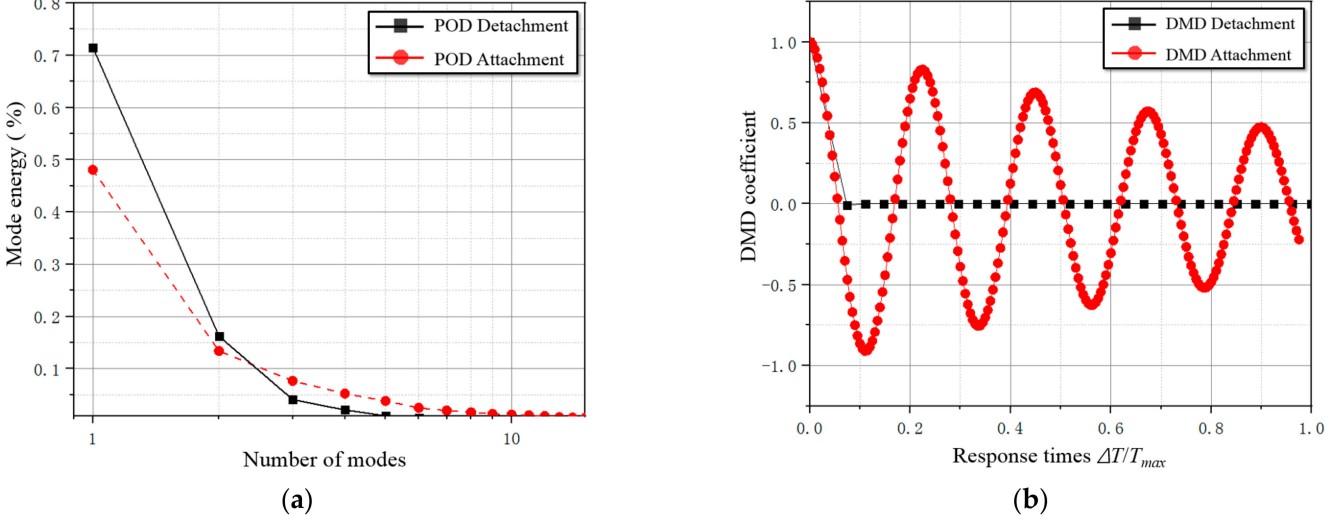

**Figure 10.** The POD results and DMD results of jet attachment and detachment conditions. (**a**) The modal energy distribution of the POD under attachment and detachment conditions; (**b**) the modal coefficient curve of the DMD under attachment and detachment conditions.

DMD is used to arrange the frequencies based on time series so that DMD can identify the stability of the flow, to a certain extent. When the modal curve is a straight line, the flow field is stable; when the modal curve is periodic, the flow field is periodic and stable; when the modal curve diverges, it indicates that the flow field is unstable; when the modal curve converges, it indicates that the flow field will be stable at a specific time. As shown in Figure 10b, the modal coefficient curve of the jet in the neutral state rapidly changes to a zero returning straight line, and the jet presents a completely stable flow state. This phenomenon indicates that the steady velocity of jet neutrality is very rapid. The modal coefficient curve of the jet attached to the wall is a pulsating curve with periodic frequency attenuation. It can be considered that the wall attachment flow is stable under steady flow and periodically stable under unsteady flow. The periodic frequency of the separation bubble structure affects the attachment flow, and changes the characteristic frequency of the flow structure under the wall−attached condition.

According to the above conclusion of jet flow stability, it can be inferred that the fundamental reason for the jet jump phenomenon is that the flow field structure inside the Coanda effect nozzle changes in the process of jet vector deflection. The stability of the neutral jet flow is steady, while the stability of the attachment flow is periodic. The vector deflection process is a process of mutual transformation between steady and periodic stable systems. The stability of the jet attached to the wall is worse than that of the jet in the neutral state, so the jet will jump when it leaves the Coanda wall.

Figure 11a shows the experimental results of oil flow on the Coanda wall at the side of the jet wall under the condition of jet wall attachment. The oil flow visualization experiment captured the shear stress distribution of the separation bubble structure on the Coanda wall. It can be seen that the separation bubble structure is a symmetrical flow structure along the $Z-$axis. The vortex−shaped structure of the shear stress line at both ends of the separation bubble indicates that there is a complex three−dimensional flow structure in the separation bubble. A pressure test on the location of the shear stress line was performed, and the test results are shown in Figure 11b. The pressure distribution on sections S1–S4 changes. At this time, the position of the lowest points ($Z/W = 0.2$, $X/L = 0.215$) and ($Z/W = 0.8$, $X/L = 0.215$) of the pressure coefficient in section S3 correspond to the core position of the vortex shear stress line. Therefore, the pulsed jet actuator is placed in the center of the vortex structure to change the flow structure in the field.

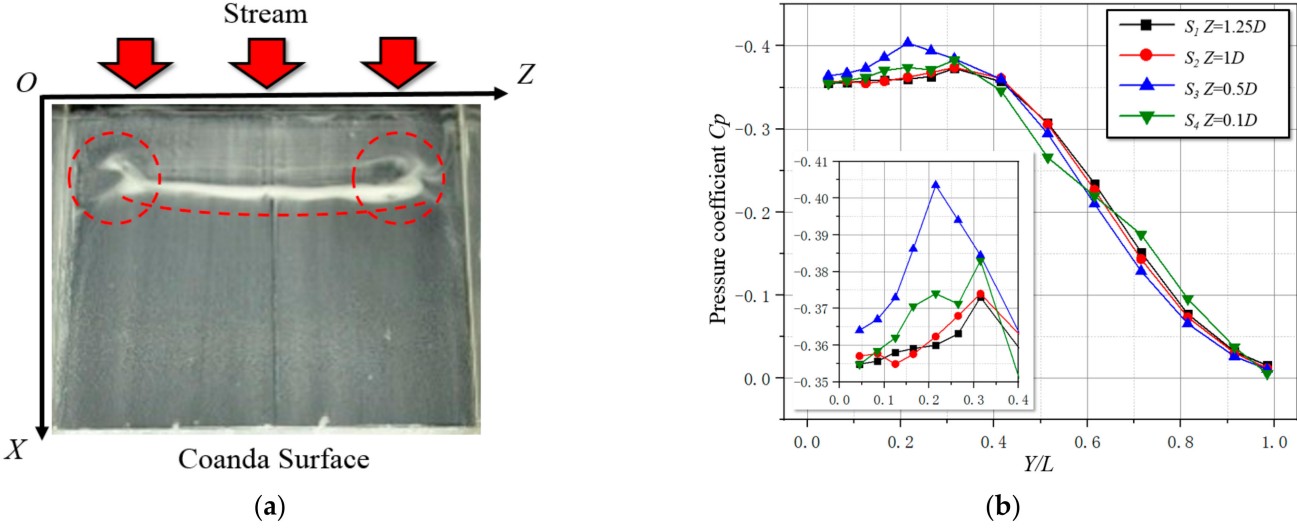

(**a**)    (**b**)

**Figure 11.** Experimental results of oil flow and pressure measurement under the condition of the jet attachment. (**a**) Results of oil flow experiments on the surface of the Coanda wall at the side of the jet attachment; (**b**) results of mean pressure measurements on the surface of the Coanda wall at the side of the jet attachment.

### 3.3. Active Flow Control Frequency Selection

The transient PIV flow field under the jet detachment process condition is shown in Figure 12; when the frame number = 10 and 30, the main jet attaches to the wall. The vortex structure produced by the outer shear layer near the side of the separation bubble structure is significant.

When the frame number = 50, the distance between the main jet and the wall increases. It is evident that the vorticity of the separation bubble structure begins to decrease. At this time, the separation bubble structure begins to move towards the trailing edge of the vectoring nozzle, and the tail of the main jet begins to leave the lower wall of the vectoring nozzle.

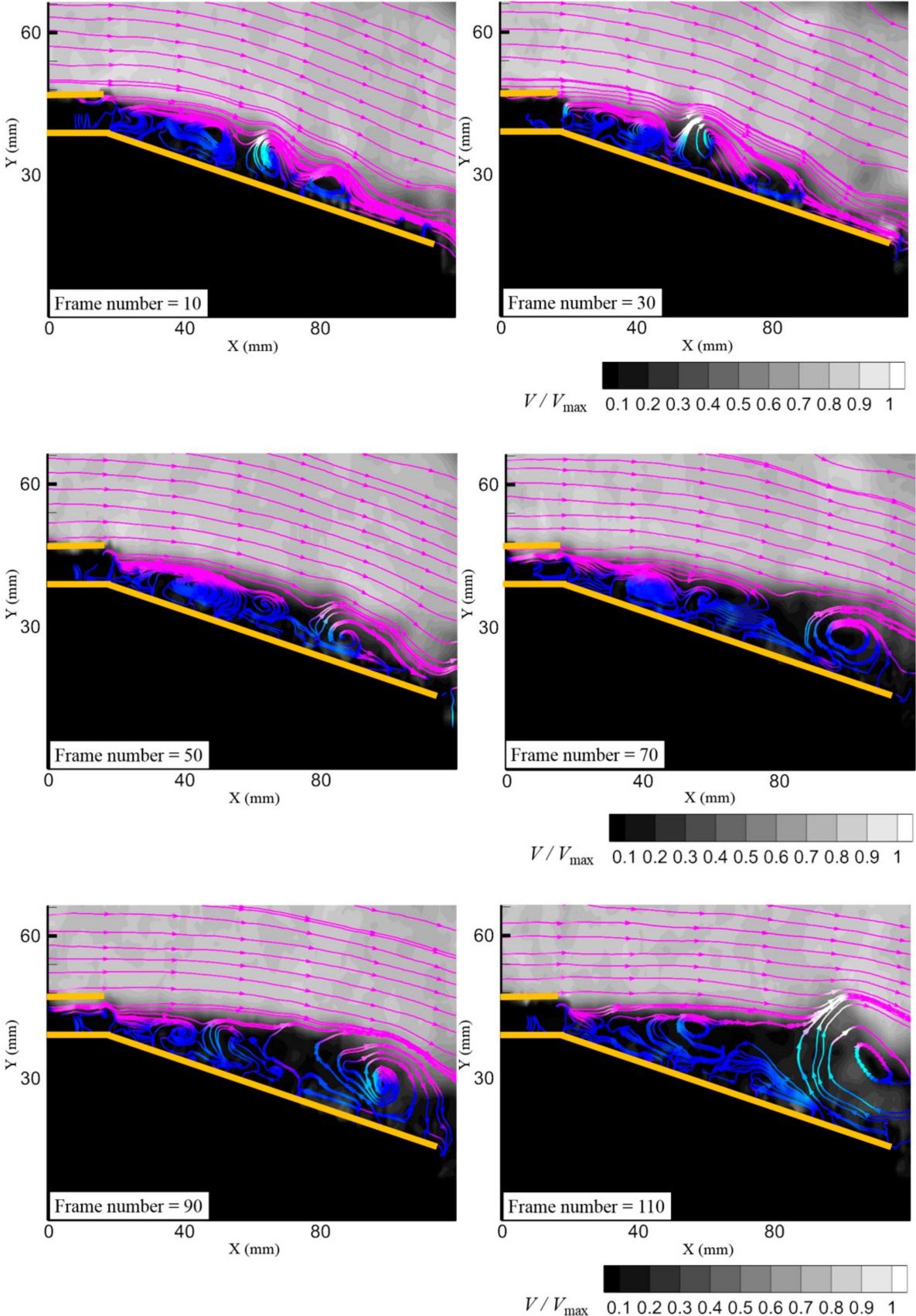

**Figure 12.** Transient PIV velocity nephogram flow field structure under the jet detachment condition (velocity streamline using RBG color).

When the frame number = 70, the vortex structure in the separation bubble area on the wall side has disappeared, the entrainment vortex begins to appear on the upper wall side of the vectoring nozzle, and the reverse suction vortex appears at the trailing edge of the lower wall of the vectoring nozzle.

When the frame number = 90 and 110, the vector jet is entirely neutral. It can be seen from the above description that when the separation bubble from before broke, the vector deflection angle of the jet is not apparent. When the separation bubble breaks, the jet begins to deflect rapidly. Therefore, maintaining the separation bubble structure can avoid the jumping phenomenon.

The PIV results of jet transient detachment process condition are used to construct the snapshot set (snapshot = 250). To determine the excitation frequency required for active flow control, dynamic mode decomposition was used to decompose the PIV snapshot set of the overall flow field and the separated bubble region (Region A). As shown in Figure 13a, the energy distribution of each mode of DMD is demonstrated. In the DMD of the global watershed, the energy proportion of the first five modes is 67.11%, 5.38%, 2.66%, 2.41%, and 2.09%, respectively. In the DMD of the separation bubble basin, the energy of the first five modes accounts for 75.48%, 9.07%, 2.59%, 1.52%, and 1.09%. Considering that the energy proportion of subsequent modes is less than 1%, the first five modes are further analyzed.

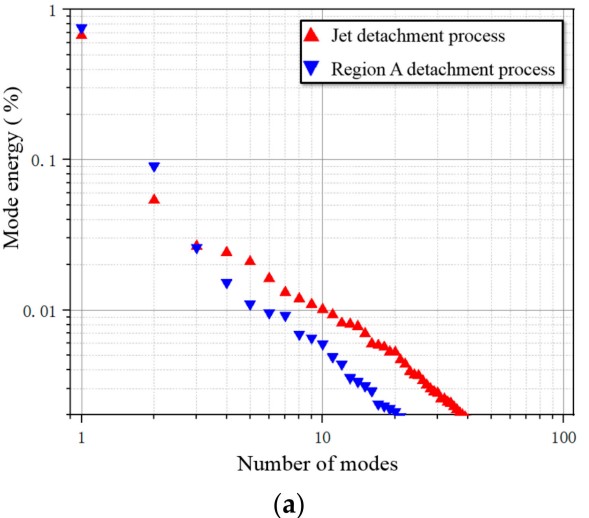
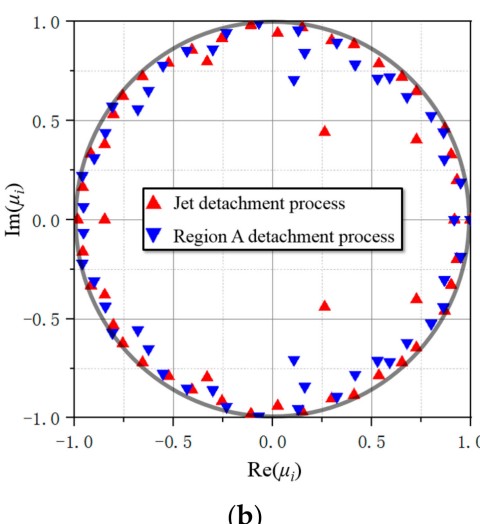

(**a**)             (**b**)

**Figure 13.** DMD results of the PIV velocity nephogram under jet transient detachment process conditions. (**a**) The modal energy distribution of DMD under the conditions of the jet detachment process; (**b**) the distribution of the real and imaginary parts of the eigenvalues on the unit circle under the conditions of the jet detachment process.

Figure 13b shows the analytical solution of the dynamic mode decomposition $\mu_i$ The distribution diagram is of the real part and imaginary part of the $\mu_i$. When the analytical solution of the DMD is distributed on the unit ring with radius one, it shows that the solved flow field is stable, and the flow field will not change with time, or has strong periodicity. When the analytical solution is outside of the unit ring, the solved flow field is divergent, and the flow in the flow field will become chaotic. When the analytical solution is in the unit ring, the solved flow field is concurrent, and the flow field will change into a more stable flow with time. The analytical solution of the transient off−wall condition of the jet is distributed in the unit ring, which shows that the off−wall process of the jet converges with time. In other words, when open, the passive secondary flow valve is on the wall side, and the injection of a passive secondary flow will not affect the flow stability of the transient jet deflection.

DMD decomposes the flow in the separation bubble region to accurately capture the required active flow control frequency. The DMD modal coefficient curve in the separation

bubble area, and the modal coefficient curve, remain convergent. As such, the DMD in the separation bubble region captures the flow mode in the separation bubble rupture process.

As shown in Figure 14, the nephogram of the DMD velocity vector of the first−order mode is shown, and the PSD of the first−order mode is 6.2 Hz. The first−order mode is the time−average result of the separation bubble region during the transient wall separation of the jet.

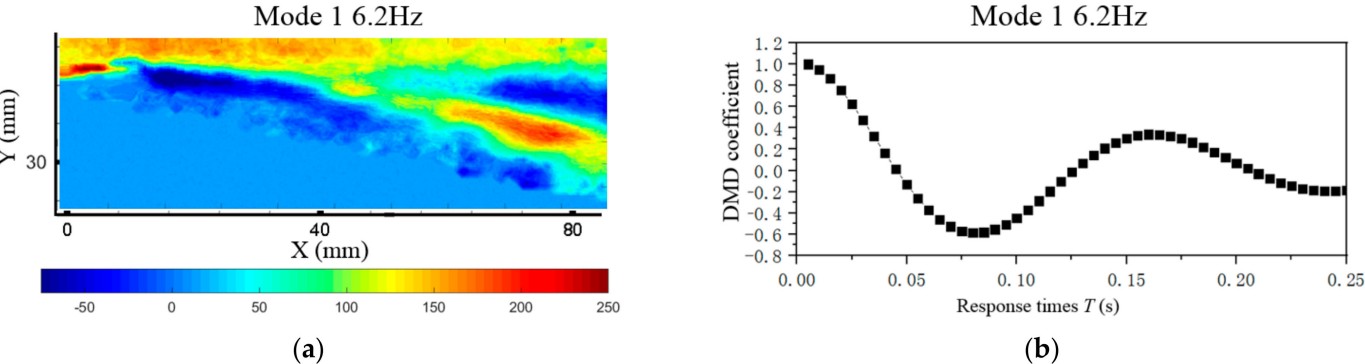

(a)  (b)

**Figure 14.** First−order mode PSD = 6.2 Hz. (**a**) Decomposition results of the PIV velocity nephogram modal DMD in the Region A; (**b**) the DMD modal coefficient curve in the Region A.

In Figure 15a the nephogram of the DMD velocity vector of third−order mode is shown, and the PSD of third−order mode is 78.16 Hz. The high−value watershed of the third−order mode is similar to the separation shedding vortex structure. In Figure 15b, the attenuation degree of the modal coefficient curve of the third−order mode is significantly higher than that of other modes, indicating that the flow region represented by the third−order mode tends to be stable at the earliest. Therefore, the third−order mode is characterized as a separation shedding vortex structure.

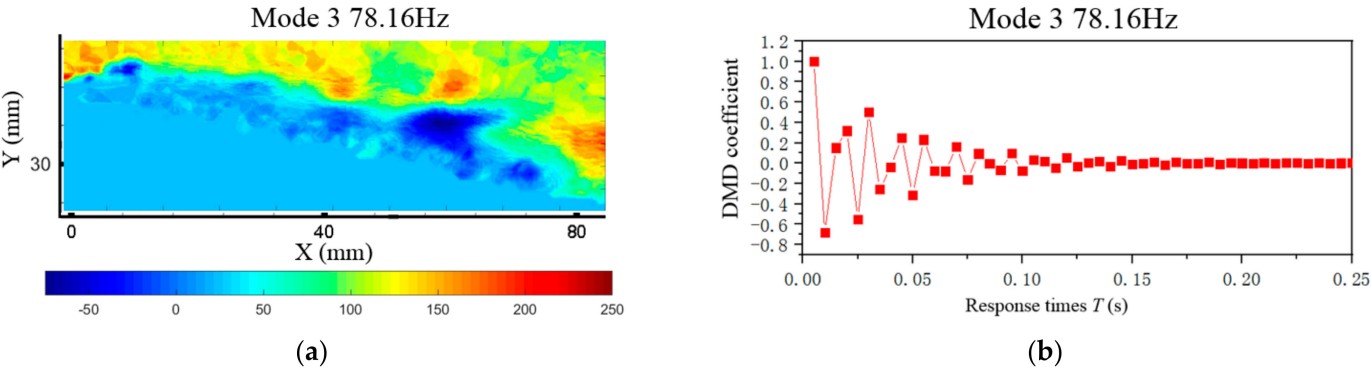

(a)  (b)

**Figure 15.** Third−order mode PSD = 78.16 Hz. (**a**) Decomposition results of the PIV velocity nephogram modal DMD in the Region A; (**b**) the DMD modal coefficient curve in the Region A.

In Figure 16a, the nephogram of the DMD velocity vector of the fourth−order mode is shown, and the PSD of fourth mode is 89.45 Hz. The high−magnitude area of the fourth−order mode is the velocity region at the trailing edge of the lower wall of the nozzle. In Figure 16b, the modal coefficient curve of the fourth mode has a specific periodic frequency, indicating that the flow represented by the fourth−order mode still exists after the separation bubble breaks. Combined with the high magnitude location, the fourth−order mode characterizes the backward suction vortex at the trailing edge of the lower wall of the nozzle. The magnitude of the velocity of the inverted vortex is shallow, which does not affect the overall flow.

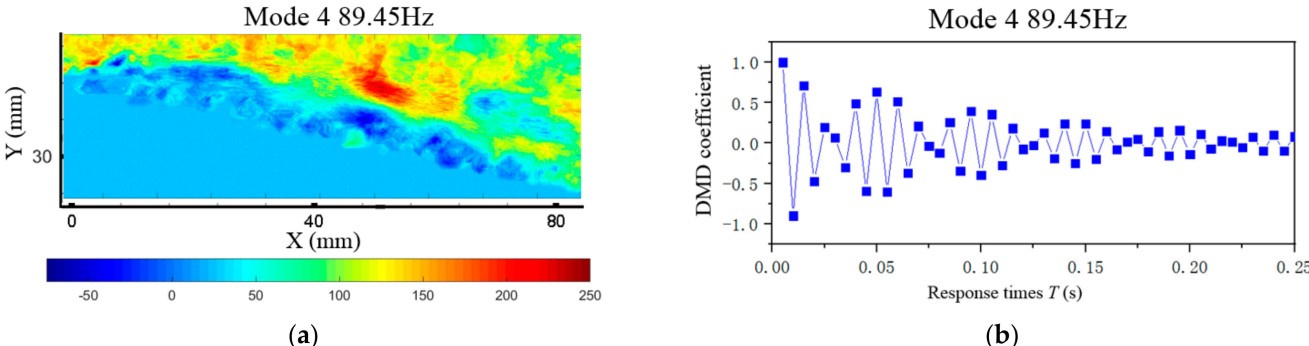

**Figure 16.** Fourth−order mode PSD = 89.45 Hz. (**a**) Decomposition results of the PIV velocity nephogram modal DMD in the Region A; (**b**) the DMD modal coefficient curve in the Region A.

As shown in Figure 17a, the nephogram of the DMD velocity vector of the fifth−order mode is shown, and the PSD of the fifth−order order mode is 97.83 Hz. The high−magnitude area of the fifth−order mode is the vortex region outside of the jet shear layer and inside the separation bubble. In Figure 17b, the attenuation degree of the modal coefficient curve of the fifth mode is linear. This linear attenuation trend is not consistent with the flow in the process of jet transient wall separation. Combined with the high magnitude location, the fifth−order mode characterizes the momentum transferred from the outer shear layer of the jet to the separation bubble. When the jet is attached to the wall in a steady state, the energy of the separation bubble structure is transferred from the momentum of the shear layer outside of the jet to the inside of the separation bubble structure. When the jet transiently leaves the wall, the momentum of the shear layer outside of the jet has no transmission medium, resulting in the gradual attenuation of the original momentum inside the separation bubble. Therefore, it can be judged that the fifth−order mode is characterized by the momentum of the shear layer outside of the jet. Combined with the above analysis, the control frequencies of the pulsed jet as 6.2 Hz and 78.16 Hz are selected. The two frequencies represent the global flow frequency and the separation bubble structure frequency in the jet separation process, respectively.

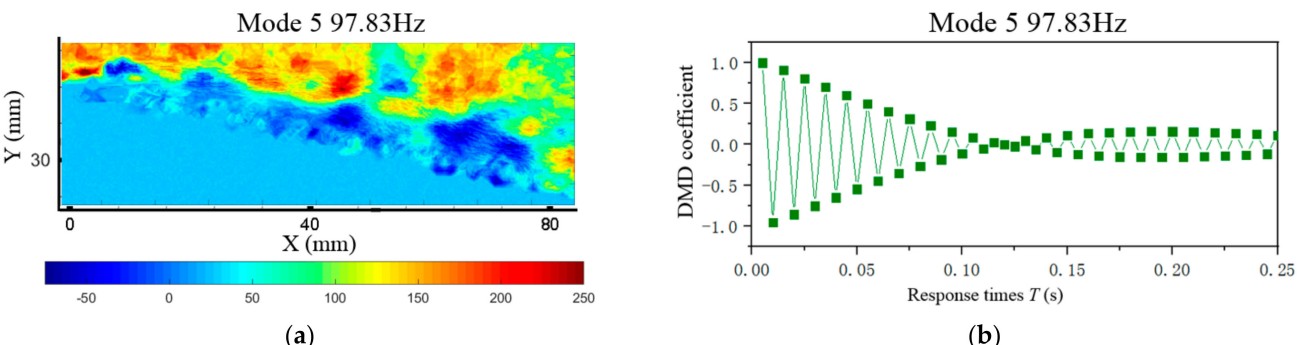

**Figure 17.** Fifth−order mode PSD = 97.83 Hz. (**a**) Decomposition results of the PIV velocity nephogram modal DMD in the Region A; (**b**) the DMD modal coefficient curve in the Region A.

### 3.4. Active Flow Control Results

Each mode frequency is used to control the jet transient vector deflection process to verify whether the control frequency of the pulsed jet is effective. The pulsed jet exciter characteristics determine the velocity of the pulsed jet.

Figure 18a shows the pressure change curve after adding the mode's PSD frequency pulsed jet in the separation bubble area. In the figure, the first−order mode 6.2 Hz and the third−order mode 78.16 Hz play the role of delay. The third−order mode characterizes the shedding vortex in the separation bubble, which shows that the characteristic frequency of the vortices in the separation bubble can inhibit the jumping phenomenon. The figure's

pressure change curves of the third, fourth and fifth modes will decrease by a specific order of magnitude after the jet completes the detachment deflection process. The third, fourth, and fifth order excitation frequencies are the same as the flow frequency inside the separation bubble, so the flow structures with different frequencies in the separation bubble resonate and generate a certain negative pressure. The negative pressure causes the pressure to drop after the jet is neutral. Figure 18b shows the variation curve of the pressure differential. The figure shows that the pulsed excitation frequency of 78.16 Hz of the third−order mode can change the high amplitude single peak curve of the original working condition into a low amplitude three−peak curve. Furthermore, the curve amplitude of the third−order mode is lower than that of the first−order mode, indicating that the pressure change controlled by the third−order mode is more linear.

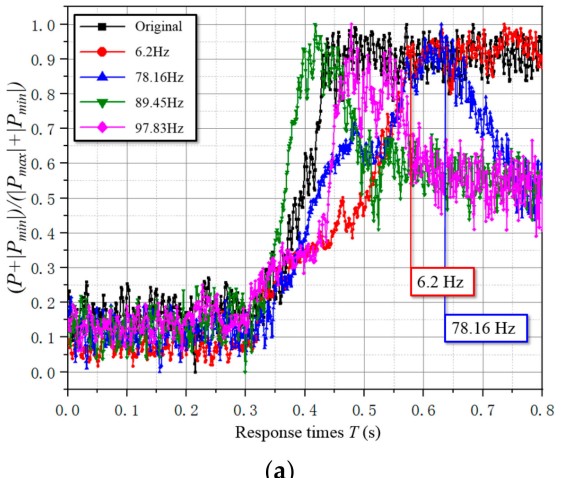
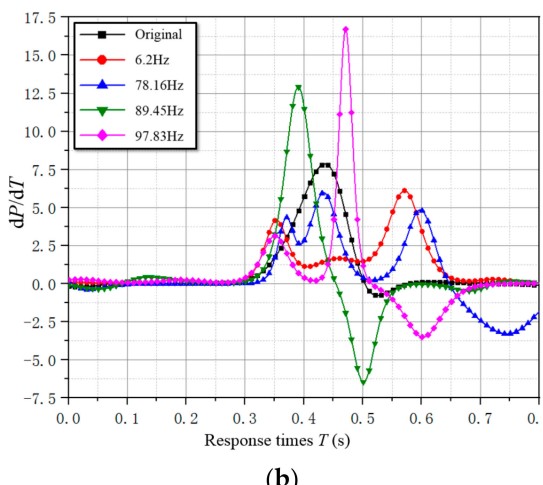

(**a**)                                                                      (**b**)

**Figure 18.** Pressure change curve under the condition of the jet transient detachment process condition. (**a**) Adding mode PSD frequency pulse jet in the separation bubble area; (**b**) pressure differential change curve in the separation bubble area.

Figure 19a compares the vector force variation curves of the first−order modal 6.2 Hz condition and the third−order modal 78.16 Hz condition. The linearity analysis is carried out for the three working conditions. The results show that the original working condition $R^2 = 0.7591$, the first−order modal 6.2 Hz working condition $R^2 = 0.9329$, and the third−order modal 78.16 Hz working condition $R^2 = 0.9637$. The linear results show that the separation bubble structure excited by the *PSD* frequency pulsed jet with third−order mode has the best inhibition effect on the jump phenomenon. Figure 19b compares the differential variation curves of a vector force. Under the original condition, the high amplitude point appears in the early stage of the jet wall separation process. At this time, the separation bubble will break in the early stage of the jet dynamic wall separation. Adding the pulsed jet with the first−order modal frequency delays the high amplitude point to the later stage of the jet wall separation process. This phenomenon shows that although the pulse jet with the first−order modal frequency can delay the rupture time of the separation bubble structure, it cannot avoid the jumping phenomenon when the separation bubble structure breaks. The absolute amplitude of the vector force differential curve decreases significantly after adding the pulsed jet with third−order modal frequency. To further clarify the action mechanism of 78.16 Hz, the pressure changing with time in the S1 section is analyzed. According to the data results of the vortex jet experiment, the vector deflection angle control equation and linear correlation square $R^2$ of the nozzle fit. The calculation formula of phenomenon correlation coefficient *R* is:

$$R = \frac{\sum_{i=1}^{n}(x_i - \bar{x})(y_i - \bar{y})}{\sqrt{\sum_{i=1}^{n}(x_i - \bar{x})^2 \sum_{i=1}^{n}(y_i - \bar{y})^2}} \tag{14}$$

where: $x_i$ and $y_i$ are the coordinates of the appropriate point $i$, respectively, and $\bar{x}$ and $\bar{y}$ are the average of $x_i$ and $y_i$, respectively.

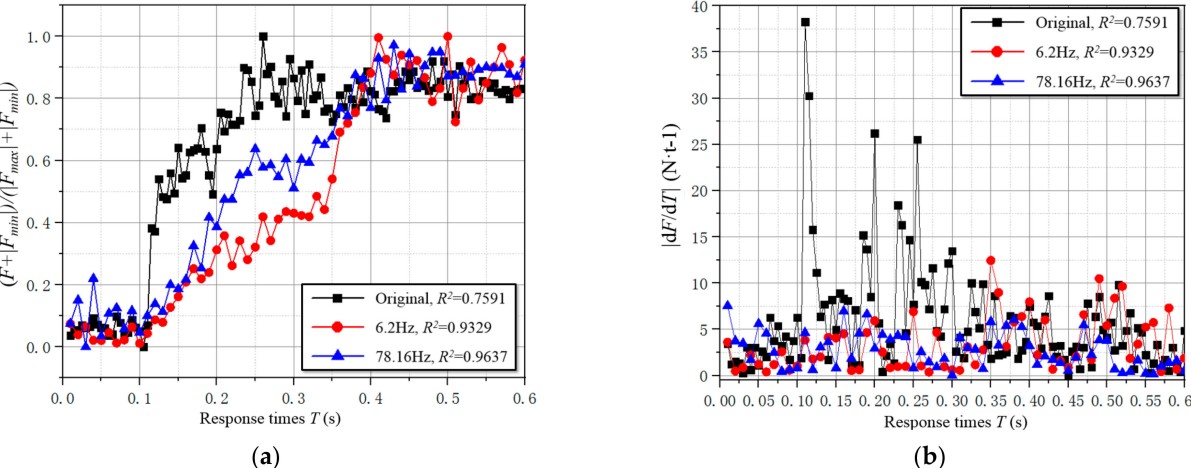

**Figure 19.** Vector force variation curve under the jet transient vector detachment process condition. (**a**) Comparison of original condition, 6.2 Hz condition, and 78.16 Hz condition; (**b**) vector force differential variation curve.

Figure 20a shows the dynamic pressure change cloud diagram in section S1 under the original working condition. When T = 0, the jet is completely attached to the wall. The blue low pressure represents the structural scale of the separation bubble. It can be observed from the figure that the separation bubble scale gradually grows with time, and disappears at a particular moment. This phenomenon describes the viscous tension generated when the main jet vector deflects the separation bubble structure. When the separation bubble bears enough tension, it will break instantly and cause a jump. Figure 20b shows the dynamic pressure change cloud diagram on S1 section under the 78.16 Hz working conditions. The figure shows that the blue low−pressure range decays linearly with time, and does not disappear instantaneously. This phenomenon indicates that the addition of a pulsed jet with the same frequency as the vortex structure in the separation bubble can enhance the structural strength of the separation bubble. After the separation bubble structure is subjected to the viscous tension generated by the vector deflection of the main jet, the separation bubble can avoid cracking, at the cost of reducing its volume.

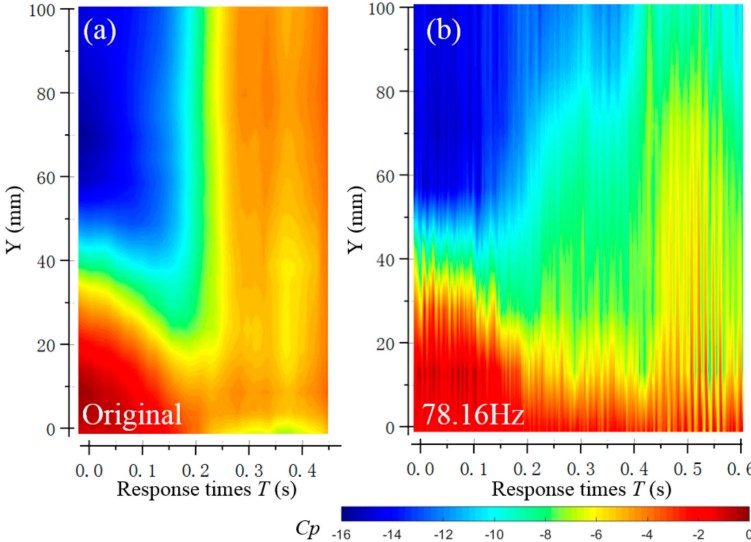

**Figure 20.** Nephogram of dynamic pressure change on S1 section. (**a**) Original working condition; (**b**) adding 78.16 Hz pulsed jet.

To sum up, the 78.16 Hz working condition has reached the hypothetical goal of this paper. The DMD method accurately obtains the characteristic frequency required in active flow control. The third−order modal results obtained by DMD in the separation bubble region characterize the vortex structure in the separation bubble structure. The active flow control method of a 78.16 Hz pulsed jet can effectively suppress the jump phenomenon and make the vector deflection angle velocity of the jet change linearly.

## 4. Conclusions

This paper investigates the vectoring deflection jumping phenomenon of the Coanda effect nozzle experimentally and numerically. The PIV technique is characterized the dynamic characteristics of jet deflection flow. According to the force measurement, pressure measurement, and DMD analysis of PIV measurement results, the phenomenon of jet deflection jump and the evolution law of vortex in a dynamic jet deflection detachment condition are investigated and discussed. Finally, according to the control object, control position, and control frequency obtained from the analysis results, the flow field's active flow control is carried out. According to the experimental model and conditions, the results are as follows:

The jumping phenomenon in the Coanda effect nozzle is caused by the different stability of the jet detachment condition and the jet attachment condition. The stability of detachment is steady, while the stability of the attachment flow is periodic. The vector deflection process is a process of mutual transformation between steady and periodic stable systems. The stability of the jet attachment is worse than that of the jet detachment state, so the jumping phenomenon will occur when the jet leaves the Coanda wall.

The DMD method can accurately analyze the jumping phenomenon in the unsteady flow field under the working conditions of this paper. The DMD method can extract the dominant frequency and the corresponding flow field modal structure in the unsteady flow field. The modal stability analysis of PIV results can accurately extract the shedding characteristic frequency of separated bubble structures. The DMD method provides a technical means for analyzing the flow field structure of the unsteady flow.

In the experimental conditions of this paper, by injecting a 78.16 Hz pulsed jet into the Coanda wall, the jumping phenomenon can be effectively suppressed, and the angular velocity of the jet vector changes linearly. The pulsed jet injection control position can effectively enhance the strength of the separation bubble structure and avoid the sudden rupture of the separation bubble structure. This active flow control method can provide a technical means for designing a new fluid thrust vectoring nozzle, by finding the characteristic frequency form of flow structure in the unsteady flow field.

**Author Contributions:** Funding acquisition, Y.G.; Writing—original draft, S.C. All authors have read and agreed to the published version of the manuscript.

**Funding:** This work was supported by A Project Funded by the Priority Academic Program Development of Jiangsu Higher Education Institutions, the National Natural Science Foundation of China, No. 11972017.

**Conflicts of Interest:** The authors declare no conflict of interest.

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
