# Peer review of "Experimental Investigation on Jet Vector Deflection Jumping Phenomenon of Coanda Effect Nozzle"

_applsci, doi:10.3390/app12157567_

Round 1

Reviewer 1 Report

The manuscript is very interesting ,the research was very well developed but some methodological issues have to be improved.

1 the abstract has to be rewritten stressing on the innovative aspects and how  could be developed the concepts presented in the present paper.

2 The introduction has to be restructured. The state of the art has to be improved ,as well as  the original contributions of the author

3Conclusions have to be more clear presented ,as well as the future implementation of the study;

4The English(Style and Grammar ) has to be improved

Reviewer 2 Report

·         The abstract of the article is like an introduction. The outcomes of research should be clearly stated in this research. Furthermore, it should mention the methodology and some important or innovative measuring devices or features.

·         This article suffers from a lack of references. Especially about the basics (Section 2), numerical methods description and formula (2.3 and 2.4), and the reported phenomena in the result section.

·         To verify whether the number of snapshots is sufficient or not, draw the convergence history diagram for POD and DMD modes.

·         Please more elaborate on the pulsed d jet

·         Add more specifications for section 2.2: no. of pressure taps, velocity sensors, and information about the device precision.

·         Please more elaborate on errors, and the ranges (systematic and random).

·         Add more references for formulas that are used. For the DMD formula, you can cite https://doi.org/10.1016/j.ijheatfluidflow.2020.108540 which clearly described the relations.

There are many Typo errors, please read the text carefully and follow the same format throughout the whole text. Some of the examples:

·         From lines 34 to 99, there is just one paragraph?

·         There is a typo in line 105: Group to solve and analyze

·         Section 2.1 needs some references for the text and Fig. 1

·         Fig. 2 and Fig. 3 are out of the center of the page.

·         In line 179: the minimum exposure time is 1.05 microsecond?

·         Line 253: in Diffusion, D should not be capital

·         Line 265: Theta C, C should be subscript.

Reviewer 3 Report

This paper considered new topic of research and represents significant interest to the readers. The material is presented in consistent fashion and attentive readers can follow the presentation. In addition to the description of Coanda effect in nozzles and presentation of the experiments, the authors provide a way on how to control the effect, so that there is no unexpected jumps in the flow. 

Some questions that authors should address:

*). In experiment width of nozzle is W=100mm. Do the boundaries have substantial effect on the flow? Please provide more clarifications.

*). Authors consider the effect of active flow control frequencies on the flow. Do the magnitudes of those frequencies depend on the size of the nozzle and the size of Coanda surfaces? 

Some minor editorial comments:

*). line 42, "studied" should be in the lower case.

*). line 45, "established" should be in the lower case.

*). line 106, "Sirovich" should be capitalized.

*). line 107, "to study" lower case.

*). line 108, "and" lower case.

*). line 338, should it be Fig 9a?

*). line 345, should it be Fig 9b?

*). line 388, should it be Figure 11(b)?

*). line 405-407, rephrase the sentence.

*). line 540, what "he"? Likely "The".

This reviewer recommends paper for the publication subject that the authors would address all the comments.

Round 2

Reviewer 1 Report

The manuscript could be published based on the improvements introduced by the authors.

Author Response

Thank you for your comments and suggestions, which greatly help us improve the paper. 

With best regards.

Reviewer 2 Report

It needs a format check.

Author Response

Thank you for your comments and suggestions, which greatly help us improve the paper. 

The format of the article has been checked and revised.

With best regards.